# Targeting E-selectin to Tackle Cancer Using Uproleselan

**DOI:** 10.3390/cancers13020335

**Published:** 2021-01-18

**Authors:** Barbara Muz, Anas Abdelghafer, Matea Markovic, Jessica Yavner, Anupama Melam, Noha Nabil Salama, Abdel Kareem Azab

**Affiliations:** 1Department of Radiation Oncology, Cancer Biology Division, Washington University in St. Louis School of Medicine, St. Louis, MO 63108, USA; bmuz@wustl.edu (B.M.); Anas.Abdelghafer@stlcop.edu (A.A.); Matea.Markovic@stlcop.edu (M.M.); jyavner@wustl.edu (J.Y.); anupama.melam@wustl.edu (A.M.); 2Department of Pharmaceutical and Administrative Sciences, St. Louis College of Pharmacy, University of Health Sciences and Pharmacy in St. Louis, St. Louis, MO 63110, USA; Noha.Salama@stlcop.edu; 3Department of Pharmaceutics and Industrial Pharmacy, Faculty of Pharmacy, Cairo University, Cairo 11562, Egypt

**Keywords:** selectins, E-selectin, uproleselan, cancer

## Abstract

**Simple Summary:**

This review focuses on eradicating cancer by targeting a surface protein expressed on the endothelium—E-selectin—with a novel drug, uproleselan (GMI-1271). Blocking E-selectin in the tumor microenvironment acts on multiple levels; uproleselan was shown (i) to inhibit cancer cell tethering, rolling and extravasating, i.e., cancer dissemination, (ii) to reduce adhesion and lose stem cell-like properties, (iii) to mobilize cancer cells to circulation where they are more susceptible to chemotherapy, which altogether contributes (iv) to overcome drug resistance. Uproleselan has been tested effective in leukemia, myeloma, pancreatic, colon and breast cancer cells, all of which can be found in the bone marrow as a primary or as a metastatic tumor site. In addition, uproleselan has a good safety profile in patients. It improves the efficacy of chemotherapy, reduces side effects such as neutropenia, intestinal mucositis and infections, and extends overall survival.

**Abstract:**

E-selectin is a vascular adhesion molecule expressed mainly on endothelium, and its primary role is to facilitate leukocyte cell trafficking by recognizing ligand surface proteins. E-selectin gained a new role since it was demonstrated to be involved in cancer cell trafficking, stem-like properties and therapy resistance. Therefore, being expressed in the tumor microenvironment, E-selectin can potentially be used to eradicate cancer. Uproleselan (also known as GMI-1271), a specific E-selectin antagonist, has been tested on leukemia, myeloma, pancreatic, colon and breast cancer cells, most of which involve the bone marrow as a primary or as a metastatic tumor site. This novel therapy disrupts the tumor microenvironment by affecting the two main steps of metastasis—extravasation and adhesion—thus blocking E-selectin reduces tumor dissemination. Additionally, uproleselan mobilized cancer cells from the protective vascular niche into the circulation, making them more susceptible to chemotherapy. Several preclinical and clinical studies summarized herein demonstrate that uproleselan has favorable safety and pharmacokinetics and is a tumor microenvironment-disrupting agent that improves the efficacy of chemotherapy, reduces side effects such as neutropenia, intestinal mucositis and infections, and extends overall survival. This review highlights the critical contribution of E-selectin and its specific antagonist, uproleselan, in the regulation of cancer growth, dissemination, and drug resistance in the context of the bone marrow microenvironment.

## 1. Introduction

E-selectin, a vascular adhesion molecule, plays a pivotal role in cell trafficking in both physiological and pathophysiological conditions. It is involved in extravasation, homing, adhesion, proliferation, stemness/cell dormancy, and drug resistance of leukocytes, hematopoietic stem cells (HSCs), and cancer cells. E-selectin is a potentially promising target for several therapeutic and medical imaging applications due to its overexpression in tissues affected by inflammation, infection, or malignancy.

E-selectin plays an important role in the interaction of cancer cells with the bone marrow (BM) microvasculature; hence, impeding these interactions not only blocks cell tethering, rolling and extravasating but also mobilizes cancer cells to circulation where they are more susceptible to chemotherapy [1]. One of the current methods used to eradicate cancer cells is to cause programmed cell death, also known as anoikis, which occurs in adhesion-dependent cells when they are forced to detach from the environment and the surrounding extracellular matrix (ECM) or are prevented from homing into a new protective BM niche [2]. A number of newly developed drugs aim to cause this programmed cell death through the disruption of the tumor microenvironment (TME) in order to target the cell–cell and cell–ECM interactions by targeting E-selectin on the supporting cells such as endothelial cells [3,4]. Moreover, there is increasing evidence showing that immune cell accumulation in the tumor as a response to chemotherapy contributes to tumor survival, less efficacious therapy, and adverse clinical events [5]. Therefore, another strategy to the improved therapeutic effect of chemotherapy is by blocking E-selectin-mediated infiltration of immune cells into tumors, as demonstrated in the breast cancer model [6].

This review focuses on the novel glycomimetic E-selectin antagonist, uproleselan (GMI-1271; GMI-1687), as an adjuvant cancer therapy. Preclinical studies demonstrated that uproleselan disrupts the interaction between the BM microenvironment and cancer cells, including leukemia, myeloma, colon, prostate, pancreatic and breast cancer cells. The results of blocking E-selectin with uproleselan were determined in vitro—where it reduced adhesion, chemotaxis, trans-endothelial migration and stroma-induced drug resistance and in animal models—where it induced stem and cancer cell mobilization from the BM to circulation and resensitized cancer cells to chemotherapies. Supporting evidence demonstrates that combination treatment with uproleselan reduced multiple myeloma (MM) resistance to carfilzomib and lenalidomide, as well as acute myeloid leukemia (AML) to cytarabine, and enhanced their therapeutic effects demonstrated by reduced tumor growth and prolonged mice survival. Moreover, uproleselan has been successfully used in clinical trials to treat patients with AML and demonstrated improved efficacy of chemotherapy and reduction of side effects such as neutropenia and infections. The trials on MM are undergoing. Based on the promising preclinical and clinical findings, targeting E-selectin has clear potential as an adjuvant cancer therapy.

## 2. The Role of E-Selectin in Cancer Pathophysiology

### 2.1. Inhibition of Selectins as a Therapeutic Strategy in Cancer

There are 3 types of selectins with a distinctive tissue expression—E-(endothelium), L-(leukocytes), and P-(platelets) selectin [1]. Selectins are major cell-surface adhesion molecules that serve as biologic brakes, rapidly decelerating leukocytes as they tether and roll on the endothelium [1,7]. Despite the low binding affinity between selectins and their ligands, it is crucial for the leukocytes to reach their destination. Following the rolling step, chemoattractant-activated leukocytes further increase the affinity to the integrins (such as VLA4) [8,9,10,11,12], then squeeze between the endothelial cells (ECs) and extravasate into specific tissues [13]. However, not only leukocytes cell trafficking is regulated by selectins, but also cancer cell adhesion, chemotaxis, stemness of the HSCs and cancer (stem) cells, and the response to anticancer drugs.

Disrupting the interaction between tumor cells and the endothelium and the TME affects cancer cell dissemination and sensitization to therapy—this can be achieved by blocking selectins [14,15,16,17,18,19,20,21,22,23]. The blockade of selectins allows for the mobilization of cancer cells, causing anoikis, which further increases their sensitivity and thus enhances the efficacy of chemotherapy. It appears that each selectin plays a major role and/or has been investigated in certain cancer models. For instance, E-selectin has been shown to be a key receptor in leukemia [24,25], myeloma [23], as well as in solid tumors such as pancreatic [26,27,28], prostate [29,30], colon [31,32] and breast [33,34,35] cancer cells. L-selectin has been shown to be a key receptor for chronic lymphocytic leukemia [20,36]. Moreover, P-selectin has been shown to play an important role in myeloma [15,18,22,23].

### 2.2. The Role of E-selectin in Cancer Progression

E-selectin, also known as CD62E, is constitutively expressed on vascular endothelium, and in BM stromal cells [23,25]. Moreover, E-selectin is upregulated in microvasculature in the presence of tumors that commonly metastasize to the bone marrow. There is a number of E-selectin ligands that are expressed on migrating cancer cells (Table 1) including E-selectin ligand (ESL-1) [37], L-selectin (CD62L) [38], P-selectin glycoprotein ligand-1 (PSGL-1, CD162) [15,18,22,39,40,41,42], CD43 [43,44], homing cell adhesion molecule 1 (HCAM1; CD44) [35,42,45,46], death receptor 3 (DR-3) [31,32] and cutaneous lymphocyte-associated antigen (CLA) [45,46].

Frequently, overexpression of functional cancer surface proteins serves as a biomarker for cancer progression and patients’ response to treatment. For instance, recent evidence suggested that CLA can play such a role in AML [17] and MM [45]. Chien et al. examined CLA expression in almost 90 AML patient samples from the peripheral blood and the BM and found a 4-fold higher expression for relapsed/refractory patients than for newly diagnosed AML patients [17]. These results were in line with increased CLA expression in cancerous plasma cells from relapsed/refractory patients compared to newly diagnosed MM patients [15,47,48]. Moreover, it was shown that CLA was increased in hypoxic MM cells, indicating the progression of MM to more advanced stages. In the mouse model, CLA^high^ MM cells were more aggressive, metastasized faster facilitating tumorigenesis, and contributed to bortezomib-mediated resistance in vivo that was reversed by blocking E-selectin [45,46]. It was also demonstrated that MM cell rolling on E-selectin in vitro was proportional to CLA levels [45]. Furthermore, circulating tumor cells were more CLA positive in relapsed MM patients than in the one isolated from the BM [45], indicating more invasive and metastatic cancer cells. These results imply that CLA undergoes dynamic changes with cancer growth and metastasis, its expression was unfavorable and correlated with worse prognosis and thus could be a potential biomarker of tumor progression and a prognostic factor of drug resistance development.

### 2.3. Signaling Pathways Regulated by E-Selectin

Some of the signaling pathways involved in E-selectin-mediated cancer functions were shown to include p38 and extracellular signal-regulated kinases (ERK)/mitogen-activated protein kinases (MAPK) ERK/MAPK), phosphatidylinositol 3-kinases (PI3K) and nuclear factor kappa-light-chain-enhancer of activated B cells (NF-kB), Wnt and Hedgehog (Table 2). The p38 and ERK MAPK pathways were shown to be involved in the migratory capabilities of colon cancer [31,32]. Esposito et al. demonstrated that the Wnt pathway is induced in breast cancer cell metastasis to the bone through activation of mesenchymal-epithelial transition (MET) and induction of stemness at the new metastatic site [54].

A mechanism of E-selectin-mediated tumor adhesion and proliferation was demonstrated to be regulated by pro-survival NF-kB and ERK signaling pathways [32,42,55,56]. Porquet et al. demonstrated that DR-3 overexpressed on HT29 and SW620 colon cancer cells interact with E-selectin, activates the antiapoptotic PI3K/NF-kB pathways, thus protects cancer cells from apoptosis [32]. Following the inhibition of PI3K and AKT pathways concurrently, the colon carcinoma cell apoptosis was increased as demonstrated by cleaved caspase-8 and caspase-3, as well as DNA fragmentation assay [32].

E-selectin is also considered a self-renewal regulator [53] by activating the cancer stemness [54,57]. Bone-homing cancer cells, especially hematological malignancies, are “hiding” in the protective and discrete E-selectin+ BM milieu that facilitates dormancy and stemness in that niche. E-selectin slows down cell division promoted by direct activation of the pro-stemness Wnt [54,57] and Hedgehog pathways (as shown in AML blasts and leukemia stem cells) [57], and pro-survival NF-kB signaling pathway [42,55,56]. It was shown that E-selectin contributes to chemotherapy resistance through cancer pro-survival (ERK/AKT), NF-kB and antiapoptotic pathways [32,42,55,56].

There is growing evidence showing that E-selectin is involved in several aspects of cancer pathophysiology:

#### 2.3.1. Cell Trafficking and Metastasis

It has been shown that cancer cells, especially hematological malignancies, use a similar system of cell trafficking to leukocytes [12,13,15,25,58]. E-selectin is involved in cancer cell trafficking and metastasis through regulating homing and engraftment [23,33,40,59,60]. Metastatic dissemination is initiated and tightly regulated by the interactions between activated E-selectin and their counter-ligands [7,29,30,40,61]. In addition, it was shown that soluble E-selectin (which sheds from the activated endothelium) contributes to CD44-expressing breast cancer cells migration and shear-resistant adhesion, facilitating leukocytes and cancer cells homing to tissues [35]. Therefore, hindering cancer cell migratory abilities by blocking E-selectin and/or their ligands is believed to hamper cancer cell extravasation and formation of new metastatic lesions in distant organs, all of which also has been scrutinized by specifically targeting E-selectin [7,15,19,22,29,62,63]. This interaction, however, may be tumor-specific since in vivo E-selectin knockout studies demonstrated that lung metastasis is not affected by the genetic deletion of E-selectin [54,55].

#### 2.3.2. Adhesion and Tumor Growth

E-selectin is a major vascular adhesion molecule [1,19]. E-selectin overexpression in cancer contributes to tumor growth due to adhesion-mediated pro-survival and antiapoptotic pathways supporting cancer proliferation. This is induced by the interaction between cancer cells with the BM microenvironment, with selectin-expressing ECs and stromal cells [15,22,32]. E-selectin is involved in cancer adhesion and adhesion-dependent cancer survival and proliferation [56]. Blocking selectins with monoclonal antibodies, by silencing the gene or by using the pan-selectin inhibitor (Rivipansel, GMI-1070), inhibited tumor adhesion dynamics and adhesion-mediated proliferation.

Additionally, Morita et al. demonstrated that E-selectin in breast cancer vasculature promotes immune cell accumulation, which facilitates tumor growth [6]. Thus, blocking E-selectin with aptamer (ESTA) significantly decreased CD45+ immune cell tumor homing in doxorubicin-treated mice, causing inhibition of tumor growth and lung metastasis. These results imply that tumor growth can be indirectly controlled by immune cell homing to the tumor through E-selectin regulation. Moreover, soluble E-selectin in the serum was described to facilitate circulating CD44-expressing cancer cells and immune cells homing to tissues, thus contributing to tumor metastasis and growth [35].

#### 2.3.3. Stemness and Self-Renewal

E-selectin is involved in HSC and cancer stemness and dormancy [25,55,57,64]. E-selectin is vital to hematopoiesis in terms of its ability to maintain steady-state expression in the BM vasculature and to retain HSCs proliferation [25]. Interestingly, the absence or blockade of E-selectin resulted in an increased proportion of quiescent HSCs, enhanced HSC survival by promoting chemoresistance [25]. Therefore, due to higher HSCs recuperation and lower BM toxicity through accelerated blood neutrophil recovery, mice with E-selectin −/− were able to survive chemotherapy 2–6-fold better than the control group; after treating both groups with antimetabolite cytotoxic 5-fluorouracil (5-FU), E-selectin −/− mice survived over 140 days while wild-type mice only survived about 48 days [57,65].

BM is hijacked by cancer cells explicitly metastasizing to the bone and utilizing this E-selectin-rich environment to become quiescent and stem-cell-like [54]. On top of that, it is facilitated by physoxia (low physiological oxygenation) present in the BM, which in the presence of the growing and expanding tumor drops, even more, contributing to hypoxic conditions mediating further stemness and drug resistance [58,64]. The presence of cancer cells in the BM is unnatural and contributes to a stressful and inflammatory environment, topped by the overexpression of E-selectin in the microvasculature [23,57,65]. Since chemotherapeutics mainly kill rapidly dividing cells, BM acts as a shield for the dormant cancer stem cells protecting them from killing [25,55,59].

#### 2.3.4. Drug Resistance

E-selectin is also involved in cancer drug resistance [15,16,66,67]. It was shown in MM and leukemia that cancer cells are protected from cytotoxic drugs due to the cell interaction with the BM vasculature inducing pro-survival signals, thus promoting cancer progression [59]. It was shown that leukemic cells with a stronger ability to bind E-selectin were 12-fold more resistant to chemotherapy in the AML mouse model [16]. In addition, gaining adhesion properties by cancer cells in suspension due to cytotoxic drug exposure, upregulated ligands and/or receptors and, as a result, conveyed drug resistance [60]. For instance, cancer cells (such as MM) overexpressing the E-selectin ligand, such as CLA, were more aggressive and more resistant to proteasome inhibitors, including bortezomib [23,45]. In addition, inhibition of these interactions using the pan-selectin inhibitor Rivipansel (GMI-1070) reversed the adhesion-mediated drug resistance induced by ECs and BM stroma in preclinical models through the sensitization of MM cells to bortezomib, which improved survival of the MM-bearing mice [15]. However, poor pharmacokinetics and a short half-life of Rivipansel requires administration of high concentration, making it inconvenient for patients. Therefore, there is a need for a selectin-specific inhibitor with better pharmacokinetics. With growing evidence demonstrating that E-selectin is involved in tumor progression and recurrence through regulation of metastasis, adhesion, stemness and drug resistance—specifically targeting E-selectin became of high interest and high importance.

## 3. Uproleselan in Cancer Therapy

The field of glycobiology in cancer has emerged since anomalous glycosylation patterns, and sialic acids and sialic acid-containing glycoconjugates associated with tumors became attractive targets for anticancer therapies. The idea started with the investigation of an enzyme called sialyltransferase (ST3 Gal-6) that mainly functions to generate E-selectin ligands. E-selectin recognizes sialylated carbohydrates/fucosylated glycoprotein ligands such as ESL-1, PSGL-1, CD44 and CLA, among others (Table 1), that are expressed on circulating leukocytes and overexpressed on cancer cells. This research provided a rationale to target E-selectin in cancer using a novel glycomimetic E-selectin antagonist, GMI-1271 (later named uproleselan), to overcome cancer spread and chemoresistance summarized in Table 3 [33].

### 3.1. Uproleselan—Chemical Structure and Properties

Uproleselan (synonym GMI-1271; chemical abstracts service (CAS) registry number: 1914993–95-5) is a small molecule glycomimetic rationally designed based on the bioactive conformation of sialyl Lea/x. Further, it is a potent and specific antagonist of E-selectin. In the target-based drug classification (PubChem.ncbi.nlm.nih.gov), uproleselan is considered a drug targeting (i) cell surface molecule and ligand, (ii) a cell adhesion molecule, and (iii) a selectin.

Uproleselan (Kd = 0.46 µM) mainly inhibits E-selectin (IC50 = 1.75 µM), but also weakly inhibits L-selectin (IC50 = 2.9 µM) and P-selectin (>10 µM). Uproleselan’s chemical formula C_60_H_108_N_3_NaO_27_ (molecular weight of 1325.70679 g/mol) is demonstrated in Figure 1.

### 3.2. The Role of Uproleselan in Cancer Therapy

#### 3.2.1. Uproleselan Inhibits Metastasis

Extravasation (egress) followed by the homing of circulating cancer cells is a crucial step of metastasis. It has been shown that uproleselan offers a promising treatment in preventing metastasis through blocking E-selectin, both in vitro (trans-endothelial migration assay) and in vivo in myeloma, pancreatic and breast cancer models [23,27,57]. The anti-homing properties of uproleselan in cancer were confirmed by inhibiting cellular interactions at every stage of cancer cell trafficking, including cancer cell retention in the blood after treating MM mouse endothelium and at the same time blocking MM cell homing to the BM and spreading the disease [23]. This specific antagonist was also shown in combination with gemcitabine to significantly reduce the frequency of metastasis of pancreatic ductal adenocarcinoma to the lymph nodes, as well as to the liver, lung and diaphragm, but did not alter primary tumor size [27]. Esposito et al. neatly demonstrated that breast cancer bone metastasis was facilitated via the E-selectin-enriched bone vascular niche, which induced MET and was inhibited by uproleselan [54].

#### 3.2.2. Uproleselan Decreases Adhesion and Activates Cancer Cell Mobilization

E-selectin performs as a gatekeeper for cancer (stem) cells from leaving or entering the BM. We and others have shown that uproleselan decreased the adhesion of cancer cells to stromal and endothelial cells in vitro [23,49,65]. In addition, static adhesion and dynamic rolling of cancer cells to E-selectin were proportional to E-selectin ligand levels and were inhibited using uproleselan [45]. As a result, blocking E-selectin activity causes de-adhesion and releases cells into the peripheral blood.

First, it was revealed by Winkler et al. that the absence of E-selectin in mice (Esel−/−) improved mobilization of HSCs, especially after granulocyte colony-stimulating factor (G-CSF) administration, which increased E-selectin expression at HSC vascular niche [25]. Then, following uproleselan administration into mice decreased cell adhesion and acted as a mobilizing agent [61].

Uproleselan combined with G-CSF enhanced mobilization of cancer cells out of the BM into the circulation much more than G-CSF alone [62]. In addition, it was shown that the E-selectin antagonist mobilized myeloma and leukemia cells from the marrow into the peripheral blood gradually within 60 min following a single injection; these cancer cells persisted in the circulation for up to 24 h and reached a ~10-fold increase at 48 h post-injection [23,45]. In comparison, a well-known CXCR4 inhibitor (AMD3100, plerixafor) rapidly mobilized tumor cells by 11-fold by 1 h, which returned to baseline within 24 h [63,71,72]. Interestingly, the most efficient cell mobilization was achieved by dual inhibition of E-selectin, and CXCR4 (using GMI-1359) mobilized leukemic cells by ~16-fold at 8 h post-injection and remained elevated even at 72 h.

One of the strategies to kill cancer cells is to expose them to systemically administered chemotherapy by anoikis [12,15,23]. Inducing de-adhesion and mobilization of cancer cells to the circulation and simultaneously not letting them home back to the marrow is achieved by the novel strategy via blocking E-selectin [22,45]. These findings demonstrate that blocking E-selectin with uproleselan mobilizes cancer cells over a long period of time, sustains the presence of tumor cells in circulation, inhibits their reentry into the BM, and thereby provides a longer window to target these cells in the circulation, sensitizing them to chemotherapy thus longer exposure to chemotherapy, and as a result significant reduction of the tumor burden [23,45].

Amongst mobilizing HSCs and cancer cells into the circulation, it was also shown that disrupting the TME with uproleselan activated the tumor-reactive and tumor-specific marrow infiltrating lymphocytes (MILs) [68]. CT26-immune mice were treated for three days with saline, G-CSF (0.125 mg/kg) or uproleselan (40 mg/kg) followed by determination of the phenotype and functional CD8+ T cells in the BM and peripheral blood 12 h after the last injection [68]. Treatment of mice with uproleselan, but not with G-CSF, led to an approximate 3–4-fold increase in naïve T cells (CD8+CD62L+CD44−) and central memory T cells (CD8+CD62L+CD44+) in peripheral blood and correlated with increased interferon gamma (IFNγ) ex vivo in response to treatment [68].

#### 3.2.3. Uproleselan Causes Maturity of Cancer Stem Cells

E-selectin was shown to be a pivotal regulator in the BM in switching between stemness/quiescence and activation/maturation of HSCs [25]. Disrupting the protective interaction between cancer cells and supportive E-selectin with uproleselan caused inhibition of quiescence through the downregulation of Wnt activity [57], and increased cell cycle and thus the maturity of cancer (stem) cells [57,65,69]. Barbier et al. demonstrated that AML-bearing mice treated with uproleselan along with chemotherapy survived longer due to chemo-sensitization of the regenerating leukemic stem cells [16].

#### 3.2.4. Uproleselan Resensitizes Cancer Cells to Therapies in Pre-Clinical Models

The main strength of utilizing multifactorial uproleselan involves its combination with other therapies. Very frequently, a single drug is not enough to successfully battle cancer and prevent tumor recurrence. Further, a plethora of evidence shows that administration of uproleselan in combination with different chemotherapies overcomes drug resistance and/or improves the efficacy of standard chemotherapy through the re-sensitization to therapies in multiple cancer models.

It was reported that in a xenotransplantation CML model, murine recipients of human CML-initiating cells treated with uproleselan and imatinib, the tyrosine kinase inhibitor, which is a standard of care in CML, further decreased the engraftment of cancer cells by decreasing their interaction time with the BM endothelium, compared to imatinib alone [53]. Uproleselan reduced adhesion of BCR-ABL1+ leukemic stem cells (LSCs) to E-selectin in the vascular niche, increased the cell cycle with simultaneous overexpression of the transcriptional regulator and protooncogene SCL/TAL1, as well as decreased CD44 expression in vitro and in vivo, thus improved eradication by imatinib [53].

Similarly, uproleselan used jointly with daunorubicin (DNR) and cytarabine (AraC) significantly improved the killing of AML cells in multiple different AML mouse models (syngeneic, xenogeneic and patient blasts), improving mice survival. Sensitization of LSCs (CD34+CD38−CD123+) to AraC chemotherapy was demonstrated by a single administration of AraC into the wild-type and Esel−/− mice, which showed that the absence of E-selectin improved LSC killing. Therefore, blocking E-selectin with uproleselan combined with AraC reversed E-selectin-induced chemoresistance to AraC and significantly decreased the number of LSCs in the femur by 95% over AraC alone [25,47,64,69].

Furthermore, it was also shown in MM mouse models (xenograft and syngeneic 5 TGM1 disseminated model) that tumors become resistant to chemotherapies due to hypoxia, adhesion to cellular and non-cellular components of the BM, and stemness [56,58,60,66,67,73,74]. Uproleselan overcame drug resistance and improved the efficacy of chemotherapies, such as proteasome inhibitors (bortezomib and carfilzomib) and IMiDs such as lenalidomide [23,47,49]. The interaction between MM cells and the TME was disrupted with uproleselan through decreasing E-selectin-mediated adhesion, stroma-induced drug resistance, chemotaxis and stemness of MM cells, which sensitized them to therapy in vitro. Additionally, uproleselan inhibited the dissemination process and therefore extended the exposure of MM cells to chemotherapies, which resulted in delayed tumor growth and prolonged mice survival [23,47,49].

One of the explanations of the improved drug efficacy that results from blocking E-selectin in the TME is the fact that uproleselan mobilizes cancer cells (AML and MM) out of the BM, causing anoikis and thus making them more susceptible to chemotherapy [4,23,69,75]. Another possible mechanism of reduced mortality of mice treated with uproleselan in combination with chemotherapy involves reducing some of the adverse effects such as BM toxicity, including neutropenia through an increased percentile of HSCs, which survived each round of treatment that facilitates blood and BM recovery and enhanced neutrophilic recovery [61,65]. Moreover, small intestine mucositis (inflammation and sloughing of the mucous membranes lining of the digestive tract) was also reduced through a reduction in the number of infiltrating inflammatory macrophages (F4/80+Ly-6C+), which normally exacerbate mucosal damage [65]. These events resulted in slowed down weight loss and eventually improved mouse survival.

The preclinical studies performed on uproleselan have revealed that blocking E-selectin affects interactions between the cancer cells and the tumor microenvironment, and thus is a valid and vital therapeutic strategy. This strategy relies on hampering the homing of already circulating tumor cells to a new metastatic niche and/or causing anoikis, a cancer death through cell de-adhesion and mobilization to the circulation, overcoming stemness and drug resistance, and further sensitizing cancer cell to chemotherapies (Figure 2).

## 4. Clinical Trials Using Uproleselan

### 4.1. Pharmacokinetics of Uproleselan

In phase I clinical trials (ClinicalTrials.gov Identifiers: NCT02168595; NCT03606447; NCT02271113), uproleselan was administered intravenously and showed favorable pharmacokinetic (PK) profiles in single doses up to 10 mg/kg in one study and up to 40 mg/kg in the second study [14,76]. Uproleselan exhibited a dose-dependent plasma levels maximum plasma-concentration (Cmax) and area under the plasma concentration time curve (AUC). Approximately two-thirds of uproleselan was excreted unchanged in the urine, and renal clearance (CLr) was on average 86 mL/min, being less than the estimated creatinine clearance (CrCl), suggesting tubular reabsorption with the apparent half-life (T_1/2_) averaging at 2.3 h [76].

### 4.2. Safety of Uproleselan (Phase I and Phase I/II Trials)

In a phase I single-dose escalation, a double-blind, placebo-controlled trial in healthy subjects was conducted to evaluate the safety, tolerability, toxicity, adverse events and PK profile of uproleselan. The drug was administered intravenously at concentrations 2 mg/kg, 5 mg/kg, 20 mg/kg and 40 mg/kg (ClinicalTrials.gov Identifiers: NCT03606447; NCT02271113). The most common adverse events were fatigue, headache, infusion site adverse events, oropharyngeal pain, presyncope and rash; however, all the above were reported as mild and unrelated to uproleselan [14,77].

In phase I/II clinical trial (ClinicalTrials.gov Identifier: NCT02306291), uproleselan was investigated in combination with chemotherapy in a total of 66 AML patients (including 44 relapses and 22 refractory patients). Increasing doses of uproleselan from 5 to 20 mg/kg for eight days were administered to patients in combination with mitoxantrone, etoposide, and cytarabine chemotherapy and monitored for adverse effects. The most frequent grade 3/4 adverse events included sepsis (18%), gastrointestinal (11%), and cardiac effects (9%). The recommended phase 2 dose (RP2 D) of 10 mg/kg was then administered to a total of 54 patients providing an optimal exposure with mucositis (2%) as the most frequent grade 3/4 adverse event. These results imply that uproleselan safety in AML is favorable even when administered with chemotherapy also in elderly AML patients [69,75].

Moreover, another phase I clinical trial (ClinicalTrials.gov Identifier: NCT02811822) for dose-escalation of uproleselan in conjunction with chemotherapy (i.e., proteasomal inhibitors such as bortezomib and carfilzomib) is currently being performed in 10 MM patients.

### 4.3. Clinical Efficacy

In a phase I/II clinical trial (ClinicalTrials.gov Identifier: NCT02306291), uproleselan was administered to 66 AML patients in combination with chemotherapy. In phase I, patients were given increasing doses of uproleselan (5–20 mg/kg) 24 h prior, every 12 h during, and 48 h post-chemotherapy, including mitoxantrone, etoposide, and cytarabine. Overall, the 60-day mortality rate of all AML patients in the study was 9%, the median overall survival was 9.2 months (95% CI, 3–12.6), and event-free survival was 12.6 months (95% CI, 9.9–NA) [78]. Currently, uproleselan is being tested in AML patients in combination with chemotherapy and compared to chemotherapy alone in phase III randomized, double-blind trial in the U.S., Australia, and Europe (ClinicalTrials.gov Identifier: NCT03616470).

In addition, in a phase I/II trial (ClinicalTrials.gov Identifier: NCT02744833), uproleselan was used as an antithrombotic treatment and tested in two patients with venous thromboembolism disease [77,79]. The symptoms of thrombosis improved, the clotting was resolved, and the associated inflammation was decreased in these patients with isolated calf-level deep vein thrombosis, with positive biological effect and improved biomarkers of coagulation, cell adhesion, and leukocyte/platelet activation [77,79].

## 5. Conclusions

Tumor cells engage specific cellular BM stroma and microvasculature and non-cellular components for tumor propagation and outgrowth, facilitating extravasation, stemness and acquisition of drug resistance. The body of work reviewed herein demonstrates that targeting E-selectin is a potential and promising adjuvant therapy to successfully disrupt the tumor microenvironment and thus kill the cancer cells more efficiently as well as reduce side effects. Recent progress in glycobiology, as well as in investigating the E-selectin—ligand-binding mechanisms, has rendered the use of specific antagonists such as uproleselan, which binds E-selectin more specifically and effectively. Uproleselan has been assessed as an adjuvant therapeutic treatment in several cancers, especially the ones relying on the protective BM milieu (leukemia and myeloma) and solid tumors metastasizing to the bone (breast, prostate, and colon cancer). This summary clearly suggests that E-selectin is a valid target, considering that it reduced cancer metastasis, detached them from a safe E-selectin-rich BM niche, mobilized cancer cells to the circulation, and overcame drug resistance by sensitizing cancer cells to chemotherapies. Both in vitro and in vivo studies supported the use of uproleselan by demonstrating significant extended mice survival rate with alleviated adverse effects such as intestinal mucositis in AML and MM-bearing mice treated with uproleselan administered together with different chemotherapies. Furthermore, completed clinical trials in AML patients using uproleselan showed high remission rates and improved overall survival with favorable safety of the drug.

Nevertheless, based on E-selectin’s important role and expression at the site of inflammation and infection, the main concern regarding targeting selectively and specifically E-selectin in the cancer tissue, and not in other inflammation sites, remains a challenge that needs to be addressed in future research.

## Figures and Tables

**Figure 1 cancers-13-00335-f001:**
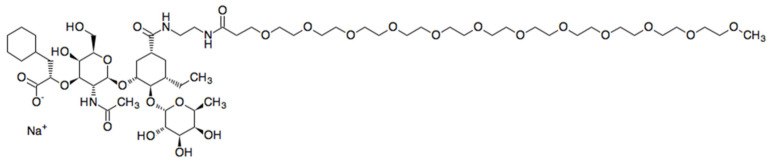
Chemical structure of uproleselan.

**Figure 2 cancers-13-00335-f002:**
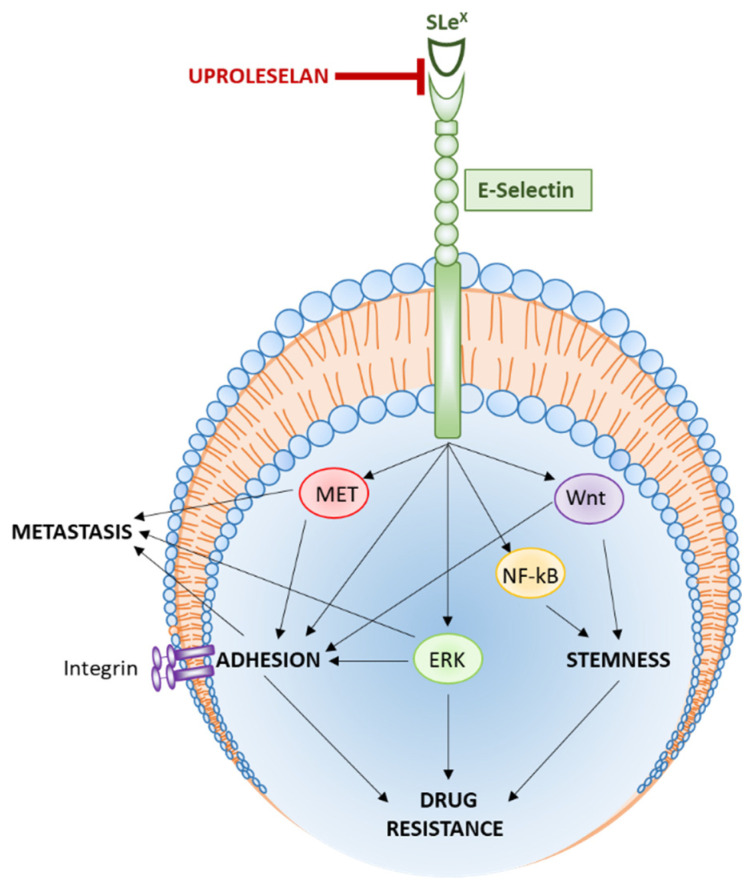
Mechanism of action of uproleselan.

**Table 1 cancers-13-00335-t001:** E-selectin ligands expressed in cancer.

E-Selectin Ligand	Full Name	Expression in Cancer	References
ESL-1	E-selectin ligand	Prostate	[47,49]
PSGL-1	P-selectin glycoprotein ligand-1; CD162	MM AML	[15,18,22] [40,42]
L-selectin	CD62L	CLL	[36]
CD43	Leukosialin, sialophorin, galactoglycoprotein	ALL DLBCL CLL Lung	[43] [48] [50] [51]
CD44	Homing cell adhesion molecule 1 (HCAM1)	AML Breast	[42,52] [35]
DR-3	Death receptor 3	Colon	[31,32]
CLA	Cutaneous lymphocyte-associated antigen	AML MM	[17] [47,49,53]

Abbreviations: MM, multiple myeloma; AML, acute myeloid leukemia; CLL, chronic lymphocytic leukemia; ALL, acute lymphocytic leukemia; DLBCL, diffuse large B-cell lymphoma.

**Table 2 cancers-13-00335-t002:** E-selectin-mediated signaling pathways.

E-Selectin-Mediated Function	Signaling Pathway	Role	Reference
Cell Trafficking and metastasis	p38 ERK/MAPK	Pro-migratory	[31] [32]
Adhesion and tumor growth	NF-kB and PI3K ERK/AKT Wnt	Pro-survival Antiapoptotic	[32] [42,55,56] [57]
Stemness and self-renewal	Wnt Hedgehog	Maintaining stemness	[54,57]
Drug resistance	ERK/AKT NF-kB	Chemoresistance Pro-survival	[32,42,55,56]

**Table 3 cancers-13-00335-t003:** Role of uproleselan in preclinical cancer models.

Role of Uproleselan in Cancer	Results	Reference
Metastasis	Prevented MM dissemination Inhibited pancreatic ductal adenocarcinoma to the lymph nodes, as well as to the liver, lung and diaphragm in combination with gemcitabine Inhibited breast cancer metastasis to the bone marrow	[23] [27] [54]
Adhesion	Decreased the adhesion of cancer cells to stromal and endothelial cells in vitro Reduced adhesion of CML leukemic stem cells to E-selectin in the vascular niche	[23,45] [53]
Mobilization	Enhanced mobilization of cancer cells out of the bone marrow into the circulation Mobilized myeloma and leukemic cells from the marrow into the peripheral blood after a single injection Activated the tumor-reactive and tumor-specific marrow infiltrating lymphocytes	[62] [23,45] [68]
Cancer stem-cell like	Inhibited cancer (stem) cell quiescence and induced cell maturation Resensitized leukemic stem cell to chemotherapy in AML-bearing mice	[57,64,65,69] [16,70]
Chemotherapy sensitization	Improved CML killing in combination with imatinib Sensitized AML in combination with daunorubicin (DNR) and cytarabine (AraC) in different mouse models (syngeneic, xenogeneic and patient blasts) Overcame MM drug resistance and improved the efficacy to proteasome inhibitors (bortezomib and carfilzomib) and IMiDs (lenalidomide)	[53] [16,62] [23,47,49,53]
Reducing adverse events	Reduced bone marrow toxicity including neutropenia, protected and increased percentile of HSCs, enhanced neutrophilic recovery, reduced small intestine mucositis by decreasing the number of infiltrating inflammatory macrophages	[16,65]

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
