# Peer review of "Targeting E-selectin to Tackle Cancer Using Uproleselan"

_cancers, 2021, doi:10.3390/cancers13020335_

Round 1

Reviewer 1 Report

This manuscript is a review of E-selectin antagonists. This manuscript is very interesting because targeting E-selectins can be a new therapeutic strategy for cancer treatment. I think this manuscript can be published with some minor revisions below.

  1. The "(ref)" on line 169 should be changed to the appropriate citation number.
  2. It is recommended to cite the following refferences about E-selectin to better demonstrate the association between E-selectin and microenvironment, metastasis, and anticancer drug resistance in solid tumors.
  • Y Morita et al. (2020) Functional Blockade of E-Selectin in Tumor-Associated Vessels Enhances Anti-Tumor Effect of Doxorubicin in Breast Cancer. Cancers. 12(3), 725.
  • SA Kang et al. (2016) The effect of soluble E-selectin on tumor progression and metastasis. BMC Cancer. 16(1), 331.

Author Response

  1. The "(ref)" on line 169 should be changed to the appropriate citation number.

ANSWER: This reference has been added as [17]. Please see Page 3 Line 109.

  1. It is recommended to cite the following references about E-selectin to better demonstrate the association between E-selectin and microenvironment, metastasis, and anticancer drug resistance in solid tumors.

Y Morita et al. (2020) Functional Blockade of E-Selectin in Tumor-Associated Vessels Enhances Anti-Tumor Effect of Doxorubicin in Breast Cancer. Cancers. 12(3), 725.

SA Kang et al. (2016) The effect of soluble E-selectin on tumor progression and metastasis. BMC Cancer. 16(1), 331.

ANSWER: We have added Morita et al (2020) reference in the following sentences:

Page 2 Line 57-61:

“There is increasing evidence showing that immune cell accumulation in the tumor as a response to chemotherapy contributes to tumor survival, less efficacious therapy, and adverse clinical events [5]. Therefore, another strategy to improved therapeutic effect of chemotherapy is by blocking E-selectin-mediated infiltration of immune cells into tumors as demonstrated in breast cancer model [6].”

Page 5 Line 201-208:

Additionally, Morita et al demonstrated that E-selectin in the breast cancer vasculature promotes immune cell accumulation, which facilitates tumor growth [6]. Thus, blocking E-selectin with aptamer (ESTA) significantly decreased CD45+ immune cell tumor homing in doxorubicin- treated mice, causing inhibition of tumor growth and lung metastasis. These results imply that tumor growth can be indirectly controlled by immune cell homing to the tumor through E-selectin regulation.

We have added Kang et al (2016) reference in the following sentences:

Page 4 Line 159-161:

“Also, it was shown that soluble E-selectin contributes to CD44-expressing breast cancer cells migration and shear-resistant adhesion, facilitating leukocytes and cancer cells homing to tissue [35]. “

Page 5 Line 206-208:

“Moreover, soluble E-selectin in the serum was described to facilitate circulating CD44-expressing cancer cells and immune cells homing to tissues, thus contributing to tumor metastasis and growth [35].”

Reviewer 2 Report

Manuscript ID: cancers-1044921

The review paper describes the function of E-selectin and its blocker uproleselan that can be used for chemotherapy. Although the article provides a useful review of tumor growth and dissemination in the bone marrow, I would like to ask several modification and reconstruction before publication.

  1. The sections 2.1, 2. 2 and 2.3 should be reconstructed. I found some redundancy, for example, the sentences in page 2, lines 89-90 and lines 96-97. The lines 85-88 can be moved to another section, for example, starting with the line 182 in page 4. This section should be described briefly the contribution of each selectin to cancers.
  2. The section 2.3.2, which includes the molecular aspects and signaling pathways of E-selectin, should come earlier than 2.3.1 and 2.3.4-5. I suggest that the authors insert a figure showing the E-selectin ligands/E-selectin interaction and their signaling pathways.
  3. The lines 110-113 in page 3 should be moved to another section, for example, starting with the line 182 in page 4.
  4. The title of 2.3.5 does not match its content. I suggest that the authors combine and organize the sections 2.3.1 and 2.3.5. I found that the lines 104-107 in page 3 and lines 161-163 in page 4 are redundant.
  5. Again, I found that the lines 189-191 in page 5 have been described in the previous section.
  6. The first sentence of the section 3.2.1 and 3.2.2 can be omitted.
  7. The section 3.2.3 is too short and rather be included in the section 3.2.2.
  8. The title of section 3.2.4 needs reconsideration. I suggest that the title includes “pre-clinical studies” or “experimental models”, so that the readers distinguish this section from the section 4.
  9. The lines 316-325 in the section 4.2 should be included in the section 3.2.4.

Author Response

 Comments:

  1. The sections 2.1, 2. 2 and 2.3 should be reconstructed. I found some redundancy, for example, the sentences in page 2, lines 89-90 and lines 96-97. The lines 85-88 can be moved to another section, for example, starting with the line 182 in page 4. This section should be described briefly the contribution of each selectin to cancers.

ANSWER: According to Reviewer #2 recommendations, we have reconstructed sections 2.1, 2.2 and 2.3, combining the first two. And we have removed the repeating sentence. We kept the lines 85-88 as is since they are regarding general Selectins and not Uproleselan.

  1. The section 2.3.2, which includes the molecular aspects and signaling pathways of E-selectin, should come earlier than 2.3.1 and 2.3.4-5. I suggest that the authors insert a figure showing the E-selectin ligands/E-selectin interaction and their signaling pathways.

ANSWER: We agree with the Reviewer and created a paragraph with subheading 2.3 “Signaling pathways regulated by E-selectin” Page 3 Line 127-148

“ Some of the signaling pathways involved in E-selectin-mediated cancer functions were shown to include p38 and ERK/MAPK, PI3K and NF-kB, Wnt and Hedgehog (Table 2). The p38 and ERK MAPK pathways were shown to be involved in migratory capabilities of colon cancer [31,32]. Esposito et al demonstrated that Wnt pathway is induced in breast cancer cell metastasis to the bone through activation of mesenchymal-epithelial transition (MET) and induction of stemness at the new metastatic site [55].

A mechanism of E-selectin-mediated tumor adhesion and proliferation was demonstrated to be regulated by pro-survival NF-B and ERK signaling pathways [32,42,56,57]. Porquet et al demonstrated that DR-3 overexpressed on HT29 and SW620 colon cancer cells interacts with E-selectin, activates the anti-apoptotic PI3K/NF-B pathways thus protects cancer cells from apoptosis [32]. Following the inhibition of PI3K and Akt pathways concurrently, the colon carcinoma cell apoptosis was increased as demonstrated by cleaved caspase-8 and caspase-3, as well as DNA fragmentation assay [32].

E-selectin is also considered a self-renewal regulator [58] through activating the cancer stemness [55,59]. Bone-homing cancer cells, especially hematological malignancies, are ‘hiding’ in the protective and discrete E-selectin+ BM milieu that facilitates dormancy and stemness in that niche. E-selectin slows down cell division promoted by direct activation of the pro-stemness Wnt [55,59] and Hedgehog pathways (as shown in AML blasts and leukemia stem cells) [59], and pro-survival NF-B signaling pathway [42,56,57].

It was shown that E-selectin contributes to chemotherapy resistance through cancer pro-survival (ERK/AKT), NF-kB and anti-apoptotic pathways [32,42,56,57]. “

We also created a Table 2 (E-selectin-mediated signaling pathways) – please see our response to Reviewer #3 recommendations.  

  1. The lines 110-113 in page 3 should be moved to another section, for example, starting with the line 182 in page 4.

ANSWER: We agree with the Reviewer and moved suggested sentences, just before introducing uproleselan.

  1. The title of 2.3.5 does not match its content. I suggest that the authors combine and organize the sections 2.3.1 and 2.3.5. I found that the lines 104-107 in page 3 and lines 161-163 in page 4 are redundant.

ANSWER: We agree with the Reviewer and moved the paragraph about CLA as a biomarker (2.3.5) to a new section 2.2, following the description of E-selectin ligand. We removed the redundant sentence and organized the E-selectin ligands into a table (also as suggested by Reviewer #3). Page 3 Line 101-122

“2.2. The role of E-selectin in cancer progression

E-selectin, also known as CD62E, is constitutively expressed on vascular endothelium, and in BM stromal cells [23,25]. Moreover, E-selectin is upregulated in microvasculature in the presence of tumors that commonly metastasize to the bone marrow. There is a number of E-selectin ligands that are expressed on migrating cancer cells (see Table 1) including E-selectin ligand (ESL-1) [37], L-selectin [38], P-selectin glycoprotein ligand-1 (PSGL-1, CD162) [15,18,22,39-42], CD43 [43,44], HCAM1 (CD44) [35,42,45,46], death receptor 3 (DR-3) [31,32] and CLA [47,48].

Frequently, overexpression of functional cancer surface proteins serves as a biomarker for cancer progression and patients’ response to treatment. For instance, recent evidence suggested that CLA can play such a role in AML [17] and MM [47]. Chien et al examined CLA expression in almost 90 AML patients samples from the peripheral blood and the BM and found a 4-fold higher expression for relapsed/refractory patients than for newly diagnosed AML patients [17]. These results were in line with increased CLA expression in myeloma cells from relapsed/refractory patients compared to newly diagnosed MM patients [15,48,49]. It was also demonstrated that MM cell rolling on E-selectin in vitro was proportional to CLA levels [47]. Moreover, it was shown that CLA was increased in hypoxic MM cells, indicating progression of MM to more advanced stages. In the mouse model, CLAhigh MM cells were more aggressive, metastasized faster facilitating tumorigenesis, and contributed to bortezomib-mediated resistance in vivo that was reversed by blocking E-selectin [47,48]. Furthermore, circulating tumor cells were more CLA positive in relapsed MM patients than in the one isolated from the BM [47], indicating more invasive and metastatic cancer cells. These results imply that CLA undergoes dynamic changes with cancer growth and metastasis, CLA expression was unfavorable and correlated with worse prognosis and thus could be a potential biomarker of tumor progression and a prognostic factor of drug resistance development.”

  1. Again, I found that the lines 189-191 in page 5 have been described in the previous section.

ANSWER: We agree with the Reviewer and removed the repetitive sentence.

  1. The first sentence of the section 3.2.1 and 3.2.2 can be omitted.

ANSWER: We agree with the Reviewer and removed the first sentences from 3.2.1 and 3.2.2.

  1. The section 3.2.3 is too short and rather be included in the section 3.2.2.

ANSWER: We expanded the paragraph about the role of uproleselan on stemness and kept it as 3.2.3 section. Page 9 Line 369-376

“E-selectin was shown to be a pivotal regulator in the bone marrow in switching between stemness/quiescence and activation/maturation of HSCs [25]. Disrupting the protective interaction between cancer cells and supportive BM by blocking E-selectin with uproleselan caused inhibition of quiescence through the downregulation of Wnt activity [59], and increased cell cycle and thus the maturity of cancer (stem) cells [55,58,77]. Barbier et al demonstrated that AML-bearing mice treated with uproleselan along with chemotherapy, survived longer due to chemo-sensitization of the regenerating leukemic stem cells [16]. “

  1. The title of section 3.2.4 needs reconsideration. I suggest that the title includes “pre-clinical studies” or “experimental models”, so that the readers distinguish this section from the section 4.

ANSWER: We altered the title of section 3.2.4 and now it reads (Page 9 Line 373): “Uproleselan Resensitizes Cancer Cells to Therapies in pre-clinical models”

  1. The lines 316-325 in the section 4.2 should be included in the section 3.2.4.

ANSWER: We agree with the Reviewer and moved the pre-clinical study results from section 4.2 to section 3.2.4 (Page 10 Line 414-416).

Reviewer 3 Report

The review “Targeting E-selectin to Tackle Cancer Using Uproleselan” explores the potential of a novel glycomimetic E-selectin antagonist, uproleselan, in blocking the interacion between tumor cells and E-selectin, which is expressed on endothelium. This has shown to contribute to the downregulation of several key features involved in tumor progression, such as cancer dissemination, as well as reduction of stem properties. Reported clinical trials highlight moderate adverse effects and promising outcomes in combination with chemotherapy. The efficacy of uproleselan as an anti-thrombotic factor is also reported. The clinical outline drawn in this review gives an intriguing cue on the role of the E-selectin antagonist.

However, I have some concerns that should be settled before publication.

Therefore, I do recommend this review to be published on Cancers after major revision.

- Figure 1. Resolution should be increased.

- To facilitate reading and give a more schematic and detailed view on the crucial role of E-selectin in cancer, authors should include, if possible, a table to highlight the main E-selectin ligands expressed on relative tumors. I am referring to chapters 2.3 and 2.3.1. Another table/ scheme that could be included regards tumor progression pathways in which E-selectin is involved. Please note that these are only suggestions. Authors can decide how to schematize this information.

- In chapter 3.2.1, authors state that “This specific antagonist was also shown in combination with gemcitabine to significantly reduce the frequency of metastasis of pancreatic ductal adenocarcinoma to the lymph nodes, as well as to the liver, lung and diaphragm…” (lines 211-213). A table outlining these results should be provided to facilitate reading.

Author Response

  1. Figure 1. Resolution should be increased.

ANSWER: We uploaded a better resolution of Figure 1.

  1. To facilitate reading and give a more schematic and detailed view on the crucial role of E-selectin in cancer, authors should include, if possible, a table to highlight the main E-selectin ligands expressed on relative tumors. I am referring to chapters 2.3 and 2.3.1. Another table/ scheme that could be included regards tumor progression pathways in which E-selectin is involved. Please note that these are only suggestions. Authors can decide how to schematize this information.

ANSWER: We agree with the Reviewer and we created a table summarizing the E-selectin ligands expressed in tumors (Table 1) as well as signaling pathways activated by E-selectin (Table 2); please see below.

Table 1 E-selectin Ligands Expressed in Cancer

E-selectin Ligand

Full name

Expression in Cancer

References

ESL-1

E-selectin Ligand

Prostate

[1,2]

PSGL-1

P-selectin glycoprotein ligand-1; CD162

MM

AML

[3-5]

[6,7]

L-selectin

CLL

[8]

CD43

leukosialin, sialophorin, galactoglycoprotein

ALL

DLBCL

CLL

Lung

[9]

[10]

[11]

[12]

CD44

HCAM1

AML

Breast

[7,13]

[14]

DR-3

Death Receptor 3

Colon

[15,16]

CLA

Cutaneous lymphocyte-associated antigen

AML

MM

[17]

[18-20]

Table 2 E-selectin-mediated signaling pathways

E-selectin mediated function

Signaling pathway

Role

Reference

Cell Trafficking & Metastasis

p38

ERK/MAPK

Pro-migratory

[15]

[16]

Adhesion & Tumor Growth

NF-kB & PI3K

ERK/AKT

Wnt

Pro-survival

Anti-apoptotic

[16]

[7,21,22]

[23]

Stemness & Self-renewal

Wnt

Hedgehog

Maintaining stemness

[23,24]

Drug Resistance

ERK/AKT

NF-kB

Chemo-resistance

Pro-survival

[7,16,21,22]

  1. In chapter 3.2.1, authors state that “This specific antagonist was also shown in combination with gemcitabine to significantly reduce the frequency of metastasis of pancreatic ductal adenocarcinoma to the lymph nodes, as well as to the liver, lung and diaphragm…” (lines 211-213). A table outlining these results should be provided to facilitate reading.

ANSWER: We agree with the Reviewer and we created a Table 3 for a better readability of the results.

Table 3 Role of uproleselan in cancer in pre-clinical models

Role of uproleselan in cancer

Results

Reference

Metastasis

·         Prevented MM dissemination

·         Inhibited pancreatic ductal adenocarcinoma to the lymph nodes, as well as to the liver, lung and diaphragm in combination with gemcitabine

·         Inhibited breast cancer metastasis to the bone marrow

[25]

[26]

[24]

Adhesion

·         Decreased the adhesion of cancer cells to stromal and endothelial cells in vitro

·         Reduced adhesion of CML leukemic stem cells to E-selectin in the vascular niche

[20,25]

[27]

Mobilization

·         Enhanced mobilization of cancer cells out of the bone marrow into the circulation

·         Mobilized myeloma and leukemic cells from the marrow into the peripheral blood after single injection

·         Activated the tumor-reactive and tumor-specific marrow infiltrating lymphocytes

[28]

[20,25]

[29]

Cancer Stem-Cell Like

·         Inhibited cancer (stem) cell quiescence and induced cell maturation

·         Resensitized leukemic stem cell to chemotherapy in AML-bearing mice

[23,24,27,28]

[21,30]

Chemotherapy Sensitization

·         Improved CML killing in combination with imatinib

·         Sensitized AML in combination with daunrubicin (DNR) and cytarbine (AraC) in different mouse models (syngeneic, xenogeneic and patient blasts)

·         Overcame MM drug resistance and improved the efficacy to proteasome inhibitors (bortezomib and carfilzomib) and IMiDs (lenalidomide)

[27]

[28,30]

[18-20,25]

Reducing Adverse Events

·         Reduced bone marrow toxicity including neutropenia, protected and increased percentile of HSCs, enhanced neutrophilic recovery, reduced small intestine mucositis by decreasing the number of infiltrating inflammatory macrophages

[30,31]

Round 2

Reviewer 3 Report

I do recommend this review for publication.